# A Branching Decoder for Set Generation

**Zixian Huang, Gengyang Xiao**
State Key Laboratory for Novel Software Technology
Nanjing University, Nanjing, China
`{zixianhuang, gyxiao}@smail.nju.edu.cn`

**Yu Gu**
The Ohio State University
Columbus, USA
`gu.826@osu.edu`

**Gong Cheng** *
State Key Laboratory for Novel Software Technology
Nanjing University, Nanjing, China
`gcheng@nju.edu.cn`

## Abstract

Generating a set of text is a common challenge for many NLP applications, for example, automatically providing multiple keyphrases for a document to facilitate user reading. Existing generative models use a sequential decoder that generates a single sequence successively, and the set generation problem is converted to sequence generation via concatenating multiple text into a long text sequence. However, the elements of a set are unordered, which makes this scheme suffer from biased or conflicting training signals. In this paper, we propose a branching decoder, which can generate a dynamic number of tokens at each time-step and branch multiple generation paths. In particular, paths are generated individually so that no order dependence is required. Moreover, multiple paths can be generated in parallel which greatly reduces the inference time. Experiments on several keyphrase generation datasets demonstrate that the branching decoder is more effective and efficient than the existing sequential decoder.

## 1 Introduction

In the past few years, the research of generative models has greatly promoted the development of AI. Pioneering studies, such as those represented by BART (Lewis et al., 2020) and T5 (Raffel et al., 2020), demonstrate that tasks with diverse structures can be seamlessly transformed into a unified text-to-text framework. This development has allowed generative models to transition from a task-specific orientation to a more versatile, general-purpose capability. The introduction of models such as GPT-3 (Brown et al., 2020) further solidified this trend within both academic and industrial spheres. Consequently, there has been a pronounced push towards delving into more nuanced tasks that generative models can address. A prime example is *set generation* (Zhang et al., 2019; Madaan et al., 2022), wherein the model is entrusted with generating a variable number of target sequences. Such functionality is imperative in contexts such as a news application displaying multiple keyphrases in a long document to assist reading (Gallina et al., 2019) or a question answering system offering multiple responses to a user query (Li et al., 2022).

Building on the foundational text-to-text paradigm, many current methods for set generation have naturally evolved to adopt the One2Seq scheme (Yuan et al., 2020; Meng et al., 2021; Madaan et al., 2022). Within this scheme, all text from the set is concatenated into a singular, extended sequence, upon which the model is trained for generation. The One2Seq scheme specifically employs a sequential decoder, generating multiple answers successively in an autoregressive fashion (see Figure 1). While several investigations explore optimizing the concatenation order (Ye et al., 2021; Cao & Zhang, 2022), they invariably retain the use of sequential decoder.

**Challenges.** However, we argue that the prevailing One2Seq approach is not optimal for set generation, presenting limitations during both the training and inference phases of model development.

---
* Corresponding author.

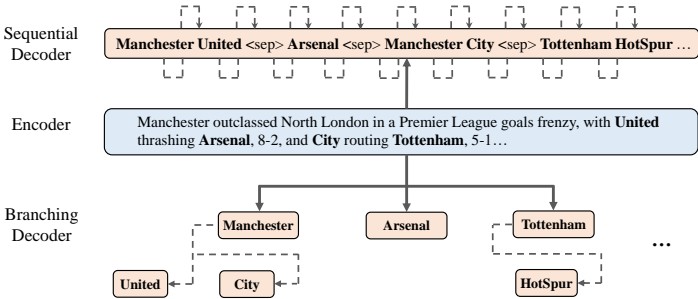

Figure 1: A comparison of traditional sequential decoder and our branching decoder for a case of set generation in keyphrase generation task.

The training phase is marked by a void in a universally recognized canonical sequence concatenation order. Distinct sequences, resulting from varied arrangements of set elements, can introduce *divergent and conflicting training signals*. This inconsistency disrupts the optimization trajectory, potentially limiting the model's performance (Meng et al., 2021; Ye et al., 2021). During inference, generating a sequence encompassing all target text proves to be considerably *time-consuming*, further underscoring the inefficiencies of the current approach (He & Tang, 2022).

**Our contributions.** To address the noted limitations inherent in the sequential decoder, we introduce a novel decoding scheme, termed **One2Branch**. This scheme is meticulously crafted to enable the generation of multiple sequences within a set concurrently. As shown in the lower segment of Figure 1, our branching decoder generates each keyphrase individually, *foregoing the need to concatenate them into a singular extended sequence*. Our novel design circumvents the challenges of ambiguous concatenation order that beleaguer the sequential decoder. In addition, by generating each sequence in the set separately, the model can capitalize on parallel computation, significantly *enhancing generation speed*. Our methodology is anchored by the branching decoder, adept at systematically delineating multiple paths from a specific input sequence, where each path uniquely signifies a sequence within the target set. A cornerstone of our contribution is the introduction of a unified framework that harmoniously integrates the ZLPR loss (Su et al., 2022) for model training (Section 3.3) and a threshold-based decoding algorithm for inference (Section 3.4). This fusion ensures that, at every decoding step, our model can selectively identify a dynamic set of tokens with logits exceeding a designated threshold (i.e., 0), facilitating the emergence of new generative branches. In a thorough evaluation spanning three representative keyphrase generation benchmarks, One2Branch unequivocally outpaces the established One2Seq techniques, yielding both augmented performance and heightened efficiency. Our methodology, seamlessly integrable with existing generative models, holds potential to redefine the paradigm for set generation using generative models.

Code: `https://github.com/nju-websoft/One2Branch`

## 2 RELATED WORK

**Text generation.** Text generation has made great progress in many aspects in recent years, such as different model architectures (Brown et al., 2020; Lewis et al., 2020; Du et al., 2021) and rich optimization techniques (Li & Liang, 2021; Goodwin et al., 2020; Houlsby et al., 2019). Existing models are usually based on the text-to-text framework (Raffel et al., 2020), i.e. the input and output of the model are unstructured text. Some works adapt the encoder to structured inputs, such as text set (Izacard & Grave, 2021), table (Liu et al., 2019), and graph (Zhao et al., 2020), but their decoder still generates an unstructured text sequence. Some works (Zhang et al., 2018; Wen et al., 2023) use multiple decoders to generate different information, but they essentially adopt a multi-view scheme to obtain a single enhanced text sequence rather than multiple text as our branching decoder. Some other works (Vijayakumar et al., 2016; Holtzman et al., 2020) study decoding strategies to obtain multiple diversified text outputs, but these unsupervised strategies need a manually specified number of generated paths. In contrast, our branching decoder is optimized to generate multiple sequences

in the training stage, and the decoding strategy in the inference stage is consistent with the behavior of training, so as to automatically determine the number of sequences generated.

**Set generation.** Many tasks in the NLP community can be regarded as set generation, such as keyphrase generation (Yuan et al., 2020; Ye et al., 2021), named entitiy recognition (Tan et al., 2021), multi-label classification (Yang et al., 2018; Cao & Zhang, 2022), and multi-answer question answering (Huang et al., 2023a;b). Early work used the One2One scheme (Meng et al., 2017) to deal with set generation, which couples each target sequence with its input sequence to form an individual training sample. However, One2One relies on manually setting a beam size to generate sequences, which results in either sacrificing recall to only generate a small number of sequences with the highest probability, or sacrificing precision to sample many sequences with a large beam number (Meng et al., 2021). Similar limitations also appear in some work based on non-autoregressive decoders (Tan et al., 2021; Sui et al., 2021) that can only generate a set of a fixed size.

Current research on set generation is mainly based on the One2Seq scheme (Yuan et al., 2020), which concatenates multiple sequences into one long sequence. However, since set is unordered, this scheme is easy to introduce order bias during training (Meng et al., 2021). The main idea to alleviate this problem is to reduce the order bias in training by finding a more reasonable concatenation order, which includes designing heuristic rules (Yang et al., 2018), using the reward feedback of reinforcement learning (Yang et al., 2019), and selecting the optimal matching between the generated sequence and the target sequence to calculate the loss (Ye et al., 2021; Cao & Zhang, 2022). Another way to enhance One2Seq is data augmentation, which adds sequences representing different concatenation orders to the training set (Madaan et al., 2022). Although these methods have been verified to alleviate the problem of order bias during One2Seq's training on some tasks, it is arguable whether a particular optimal order exists for tasks such as keyphrase generation. Compared with the above work, our One2Branch scheme not only can generate an unfixed number of sequences like One2Seq, but also is not troubled by the disorder of the set like One2Seq.

## 3 ONE2BRANCH: A BRANCHING DECODER

### 3.1 OVERVIEW

Given an input sequence $X = \mathrm{X}_1, \ldots, \mathrm{X}_l$ with $l$ tokens, the goal of set generation is to generate a target set $\mathbb{Y} = \{Y_1, \ldots, Y_n\}$ containing $n$ target sequences, where each text output $Y_i = \mathrm{Y}_{i,1}, \ldots, \mathrm{Y}_{i.\boldsymbol{m}_i}$ contains $\boldsymbol{m}_i$ tokens. A vocabulary $V = \mathrm{V}_1, \ldots, \mathrm{V}_\mu$ containing $\mu$ tokens is predefined and all generated tokens are selected from it. The operator $\mathrm{Index}(\cdot)$ is used to map tokens to their index positions in the vocabulary, i.e. $\mathrm{Index}(\mathrm{V}_i) = i$.

Figure 2 gives an example of the generation strategy of our branching decoder, containing 3 generation paths that correspond to 3 target sequences. Compared with the existing generative model using a sequential decoder, the branching decoder has two characteristics: (1) generating a dynamic number of tokens at each time-step, and (2) generating multiple target sequences in parallel.

Specifically, different from sequential decoder that can only select a fixed beam number of tokens, branching decoder uses a thresholding strategy to allow selecting a dynamic number of tokens, so one or more new generation paths can be branched at each time-step. As shown in Figure 2, branching decoder can generate 2-3-3-1 tokens respectively at time-step 1 to 4. Benefitting from the ability to dynamically branch, multiple target sequences can be generated in parallel. In Figure 2 the branching decoder can use 1-2-3-1 decoders to generate multiple target sequences in parallel at time-step 1 to 4. Compared with the traditional sequential decoder that needs to use a total of 12 time-steps (including 2 separating tokens) to generate all target sequences, the branching decoder only needs to use 4 time-steps since multiple sequences can be generated in parallel.

### 3.2 ARCHITECTURE

Our One2Branch scheme uses the classic encoder-decoder architecture, but the output of the encoder can be fed into multiple parameter-sharing decoders to generate in parallel. A stacked transformer encoder first encodes the input sequence to obtain its hidden states representation:

$$\boldsymbol{H}^{\mathrm{E}} = \mathrm{Encoder}(X)\,, \tag{1}$$

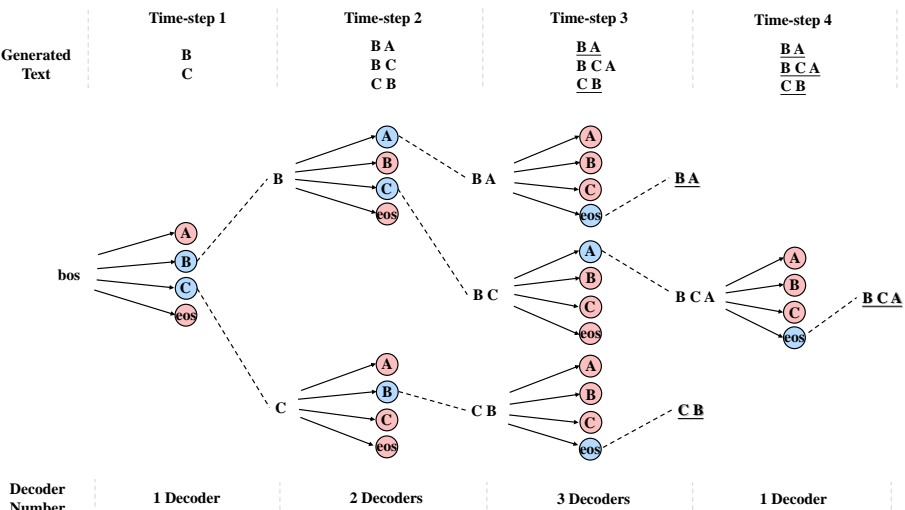

Figure 2: An example of branching decoder for set generation. It uses 0 as the threshold to select nodes (blue) at each time-step and generates the set {"B A", "B C A", "C B"}.

where $\boldsymbol{H}^{\mathrm{E}} \in \mathbb{R}^{l \times d}$ and $d$ denotes the dimension of representation.

At the $t$-th time-step, for the $i$-th generated path with $t-1$ generated tokens $P_{i,:t-1}$, the representations of the 1-st to $t$-th time-steps can be obtained as:

$$\boldsymbol{S}_{i,:t}, \boldsymbol{H}_{i,:t} = \texttt{Decoder}(P_{i,:t-1}, \boldsymbol{H}^{\mathrm{E}}), \tag{2}$$

where $\texttt{Decoder}(\cdot, \cdot)$ is an autoregressive decoder composed of stacked transformer, $\boldsymbol{S}_{i,:t} \in \mathbb{R}^{t \times \mu}$ and $\boldsymbol{H}_{i,:t} \in \mathbb{R}^{t \times d}$ contain generation scores and representations of $t$ tokens on the $i$-th path, respectively.. For the $t$-th token, its generation scores $\boldsymbol{S}_{i,t} \in \boldsymbol{S}_{i,:t}$ is a vector containing the score of each token in the predefined vocabulary $V$. For the tokens with a score greater than the threshold, they will be selected as generated tokens and each of them will branch a new path at the next time-step. In the following, we will introduce our training strategy, which enables a fixed threshold to decode a dynamic number of tokens and branch out to multiple paths to generate in parallel.

## 3.3 TRAINING

Our training strategy consists of: sharing decoder, learning threshold, and negative sequence.

**Sharing decoder.** The branching decoder should generate multiple unordered target sequences in parallel, so we adopt a sharing decoder way to train the model. As shown in Figure 3, each target sequence is independently input into a decoder with shared parameters using teacher forcing manner to obtain the generation score of each token in the vocabulary, which is implemented by Equation (2):

$$\boldsymbol{S}_i, \boldsymbol{H}_i = \texttt{Decoder}([\langle \mathrm{bos} \rangle; Y_i], \boldsymbol{H}^{\mathrm{E}}), \tag{3}$$

where $\boldsymbol{S}_i \in \mathbb{R}^{(|Y_i|+1) \times \mu}$ and $\boldsymbol{H}_i \in \mathbb{R}^{(|Y_i|+1) \times d}$ are the generation scores and representations of all the $|Y_i| + 1$ tokens on the $i$-th path, respectively, $[\cdot; \cdot]$ is the operator of concatenation, and $\langle \mathrm{bos} \rangle$ is a special token used to represent the beginning of a generated sequence.

Although each sequence is input to a decoder independently, their label sets are related to each other. As illustrated in Figure 3, when the decoder inputs are the common prefix $\langle \mathrm{bos} \rangle$ of all the target sequences, because both "B" and "C" are tokens that can be generated next, the positive label sets are $\Omega_{i,1}^{+} = \texttt{Index}(\{\text{"B"}, \text{"C"}\})$ at time-step 1 for every decoder $i$. Similarly, when the decoder inputs are the common prefix "$\langle \mathrm{bos} \rangle$ B" of the target sequences "B A" and "B C A" , the positive label sets are $\Omega_{i,2}^{+} = \texttt{Index}(\{\text{"A"}, \text{"C"}\})$ at time-step 2 for decoders $i = 1$ and $i = 2$. For each target sequence at each time-step, the negative label set is obtained by removing the positive label set from the vocabulary, i.e. $\Omega_{i,t}^{-} = \texttt{Index}(V) \backslash \Omega_{i,t}^{+}$.

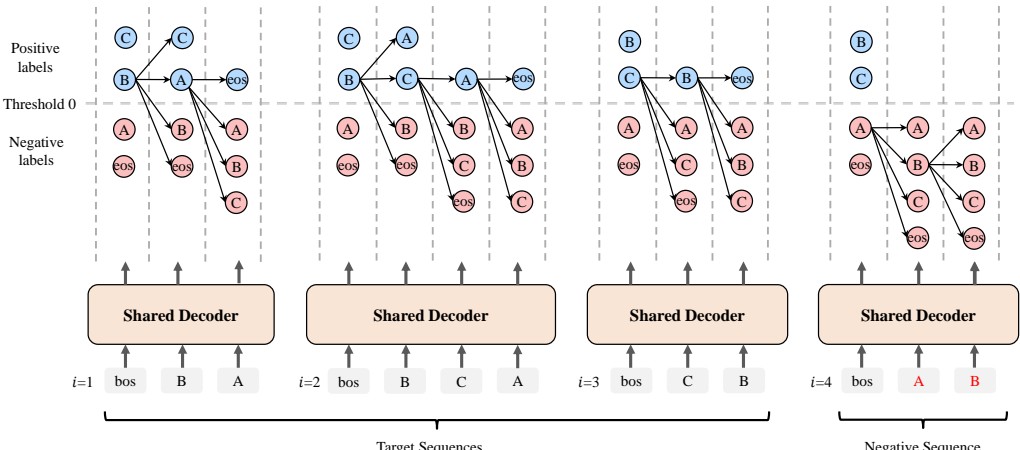

Figure 3: An example of One2Branch scheme's training strategy. For the target set {"B A", "B C A", "C B"}, One2Branch uses three decoders with shared parameters to learn from each target sequence in the teacher forcing manner. A negative sequence "A B" is also used to train. For each input token, the scores of positive tokens in the vocabulary are optimized to be greater than the threshold 0, and the scores of the remaining negative tokens are optimized to be less than 0.

If two target sequences $Y_i$ and $Y_j$ have a common prefix up to time-step $t$ but differ at time-step $t + 1$, then the positive label sets of the two sequences at time-step $t$ separately include $Y_{i,t+1}$ and $Y_{j,t+1}$. This approach of constructing labels does not fix the number of positive and negative labels, which requires the model to learn to predict a dynamic number of tokens at each time-step.

**Learning threshold.** In order to realize the selection of a dynamic number of tokens at each time-step to branch multiple paths, we expect that the model learns a threshold to qualify tokens that can be selected. Benefiting from the work of Su et al. (2022) used in multi-label classification task, we introduce the ZLPR loss to generative model to learn the threshold at 0:

$$\mathcal{L} = \log(1 + \sum_{i=1}^{n} \boldsymbol{s}_i^+) + \log(1 + \sum_{i=1}^{n} \boldsymbol{s}_i^-)$$

$$\text{where} \quad \boldsymbol{s}_i^+ = \sum_{t=1}^{\boldsymbol{m}_i} \sum_{w \in \Omega_{i,t}^+} \exp(-\boldsymbol{S}_{i,t,w}), \quad \boldsymbol{s}_i^- = \sum_{t=1}^{\boldsymbol{m}_i} \sum_{w \in \Omega_{i,t}^-} \exp(\boldsymbol{S}_{i,t,w}),$$

(4)

where $n$ is the size of the target sequences set, $\boldsymbol{m}_i$ is the length of target sequence $Y_i$, and $\boldsymbol{S}_{i,t,w} \in \mathbb{R}$ is the generation score of token $V_w$ at time-step $t$ on the $i$-th path. When using Equation (4) to supervise model training, both $\boldsymbol{s}_i^+, \boldsymbol{s}_i^- \in (0, +\infty]$ will be optimized to be minimal, meaning the scores of position tokens in $\Omega^+$ will be optimized toward positive infinity away from 0, and the scores of negative tokens in $\Omega^-$ will be optimized toward negative infinity away from 0, which enables using 0 as the threshold in the inference stage.

**Negative sequence.** Recall that a common way to train a decoder is to calculate the loss from positive sequences using teacher forcing, which often leads to exposure bias and affects model generalizability (Bengio et al., 2015). To address it, we incorporate a loss from negative sequences to supervise the model in distinguishing between positive and negative sequences.

From the first incorrect token of a negative sequence, for all the subsequent tokens, we construct their label sets $\Omega_{i,t}^+ = \varnothing$, $\Omega_{i,t}^- = \text{Index}(V)$. As Figure 3 shows, "A B" is a negative sequence; "A" and "B" in the second and third time-steps are negative tokens with empty positive label sets.

In practice, we train branching decoder in two stages. In the first stage, only target sequences are used. In the second stage, we first use the model trained in the first stage to predict on the training set and collect the incorrectly generated sequences having the highest scores as hard negatives, and then further train branching decoder with both target sequences and negative sequences.

---

**Algorithm 1** Decoding

**Input**: $X$, step$^{\max}$, $k^{\min}$, $k^{\max}$

1: $\mathbb{F}, \mathbb{C}^{F} \leftarrow \varnothing, \varnothing$. // $\mathbb{F}$: the set of finished paths. $\mathbb{C}^{F}$: the scores of $\mathbb{F}$
2: $\mathbb{U}, \mathbb{C}^{U} \leftarrow \{\langle \mathrm{bos} \rangle\}, \{0\}$ // $\mathbb{U}$: the set of unfinished paths. $\mathbb{C}^{U}$: The scores of $\mathbb{U}$.
3: $\boldsymbol{H}^{E} \leftarrow \texttt{Encoder}(X)$
4: $t \leftarrow 1$
5: **while** $\mathbb{U}$ is not empty **AND** $t \leq$ step$^{\max}$ **do**
6:      $\mathbb{B}, \mathbb{C}^{B} \leftarrow \varnothing, \varnothing$ // $\mathbb{B}$: the set of newly branched paths. $\mathbb{C}^{B}$: The scores of $\mathbb{B}$.
7:      $\boldsymbol{S}, \boldsymbol{H} = \texttt{Decoder}(\mathbb{U}, \boldsymbol{H}^{E})$ // Calculate the generation scores of all the paths in $\mathbb{U}$ in a batch.
8:      **for** $k \leftarrow 1$ to $k^{\max}$ **do**
9:          $\boldsymbol{S}_{i,t,w}, i, w \leftarrow k\text{-thLargest}(\boldsymbol{S}_{:,t})$ // Return the $k$-th largest score $\boldsymbol{S}_{i,t,w}$, and its path index $i$ and vocabulary index $w$ at time-step $t$.
10:          **if** $\boldsymbol{S}_{i,t,w} > 0$ or $k \leq k^{\min}$ **then**
11:              $\mathbb{B}.\texttt{add}([\mathbb{U}_i; \mathrm{V}_w]), \mathbb{C}^{B}.\texttt{add}(\mathbb{C}^{U}_i + \boldsymbol{S}_{i,t,w})$
12:      $\mathbb{F}, \mathbb{C}^{F} \leftarrow \texttt{EndWithEos}(\mathbb{B}, \mathbb{C}^{B}, \mathbb{F}, \mathbb{C}^{F})$ // If a path has generated the $\langle \mathrm{eos} \rangle$, it is a finished path.
13:      $\mathbb{U}, \mathbb{C}^{U} \leftarrow \mathbb{B} \backslash \mathbb{F}, \mathbb{C}^{B} \backslash \mathbb{C}^{F}$
14:      $t \leftarrow t + 1$
15: **return** $\mathbb{F}, \mathbb{C}^{F}$

---

## 3.4 DECODING

Using the above training strategy, as illustrated in Figure 2, the branching decoder can select an unfixed number of tokens with a generation score greater than 0 at each time-step in the inference phase, and dynamic branch out new paths to continue generating in parallel at the next time-step. However, this naive decoding strategy may suffer from insufficient generation—branching out too few paths, or excessive generation—branching out too many paths to fit in the memory. In order to avoid such extreme cases, we set the minimum number of explored paths $k^{\min}$ and the maximum number of generated paths $k^{\max}$, and rely on the average score of the entire path instead of the score of the current token to smooth the result.

We show the decoding algorithm of branching decoder in Algorithm 1. For an input sequence $X$, we first use Equation (1) to obtain its representation $\boldsymbol{H}^{E} \in \mathbb{R}^{l \times d}$ (line 3), and then iterate up to step$^{\max}$ time-steps to parallelly generate paths (lines 5-14). In each iteration, the representation $\boldsymbol{H}^{E}$ is repeated $|\mathbb{U}|$ times and fed to Equation (3) with the set $\mathbb{U}$ to obtain the scores of all possible generated paths $\boldsymbol{S} \in \mathbb{R}^{n \times |\mathbb{U}| \times \mu}$ (line 7). In order to avoid potential memory overflow caused by generating too many paths, we only generate new paths from the $k^{\max}$ highest-scored tokens among all $n \times \mu$ candidate tokens at time-steps $t$ (lines 8–9). Then, the candidate tokens $\mathrm{V}_w$ whose generation scores $\boldsymbol{S}_{i,t,w}$ are greater than 0 or are among the top-$k^{\min}$ are appended to path $\mathbb{U}_i$ and added to the set $\mathbb{B}$ as new paths, and their corresponding generation scores are added to set $\mathbb{C}^{B}$ (lines 10–11). The paths that end with the terminal symbol $\langle \mathrm{eos} \rangle$ are moved into the set $\mathbb{F}$, their corresponding scores are moved to $\mathbb{C}^{F}$ (line 12), and the rest will continue to generate in the next time-step (line 13). Finally, we filter out the paths whose average score is less than 0 to obtain the final generated sequences.

## 4 EXPERIMENTAL SETUP

### 4.1 DATASETS AND EVALUATION METRICS

Keyphrase generation (KG) is a classic task of set generation with rich experimental data. We selected three large-scale KG datasets as our main experimental data: **KP20k** (Meng et al., 2017), **KPTimes** (Gallina et al., 2019), and **StackEx** (Yuan et al., 2020), which are from the fields of science, news and online forums, respectively. Each dataset contains not only keyphrases that are present in a document but also those that are absent. Dataset statistics are shown in Table 1.

Following Chen et al. (2020), we used macro-averaged F1@5 and F1@M to report the generation performance of present and absent keyphrases. When the number of predicted keyphrases is below 5, F1@5 first appends incorrect keyphrases until 5 and then compares with ground-truth to calculate

Table 1: Dataset statistics. # KP, |KP|, and % Abs KP refer to the average number of keyphrases per document, the average number of words that each keyphrase contains, and the percentage of absent keyphrases, respectively. All of them are calculated over the dev set.

| Dataset | Field | # Train | # Dev | # Test | # KP | |KP| | % Abs KP |
|---------|-------|---------|-------|--------|------|------|----------|
| KP20k | Science | 509 K | 20 K | 20 K | 5.3 | 2.1 | 39.8 |
| KPTimes | News | 259 K | 10 K | 20 K | 5.0 | 2.2 | 56.4 |
| StackEx | Forum | 298 K | 16 K | 16 K | 2.7 | 1.3 | 46.5 |

F1 score defined by Yuan et al. (2020). F1@M is the version of F1@5 without appending incorrect keyphrases, which compares all predicted keyphrases with the ground-truth to compute F1 score.

## 4.2 Implementation Details

We implemented our One2Branch scheme based on the code of huggingface transformers 4.12.5 [1] and used T5 Raffel et al. (2020) as backbone. For the two stages of training, we trained 15 epochs in the first stage, and 5 epochs in the second stage. We set $\text{step}^{\max} = 20$ to ensure that $\text{step}^{\max}$ is greater than the length of all keyphrases on all dev sets.

We set $k^{\max} = 60$ to ensure that $k^{\max}$ is greater than all numbers of keyphrases on all dev set. We tuned $k^{\min}$ on each dev set from 1 to 15 to search for the largest sum of all metrics, and the best $k^{\min}$ was used on the test set. On all three datasets, the best performance was achieved with $k^{\min} = 8$.

We followed the setting of Wu et al. (2022) using batch size 64, learning rate $1e - 4$, maximum sequence length 512, and AdamW optimizer. We used three seeds $\{0, 1, 2\}$ and took the mean results. We used gradient accumulation 64, trained One2Branch based on T5-Base (223 M) on a single RTX 4090 (24 G), and trained One2Branch based on T5-Large (738 M) on two RTX 4090. For inference, we ran both base and large versions on a single RTX 4090. Unless otherwise stated, the results of our baseline methods came from the work of Wu et al. (2022).

We also implemented our One2Branch scheme based on MindSpore 2.0.

## 5 Experimental Results

### 5.1 Main Results: Comparison with Sequence Decoder

We presented the comparison of generation performance, inference throughput and GPU memory usage between the One2Branch scheme based on branching decoder and the One2Seq scheme based on sequential decoder. The implementation of One2Seq was trained with Present-Absent concatenation ordering, using a delimiter ⟨sep⟩ to concatenate the present keyphrases and the absent keyphrases, which has been seen as an effective ordering in previous works (Yuan et al., 2020; Meng et al., 2021). We experimented the throughput and GPU memory usage with batch size 1 on a single RTX 4090, and used greedy search to decode One2Seq (see results in Table 2).

**Generation performance.** One2Branch outperforms One2Seq in all F1 scores on absent keyphrase and more than half of F1 scores on present keyphrase.

For absent keyphrases, they are completely unordered, and it is difficult for One2Seq to give a reasonable concatenation order to avoid bias during training, while One2Branch is a label order agnostic model which can deal with this problem. The results in Table 2 support our motivation; compared with One2Seq, One2Branch improves by up to 3.5, 11.6, and 10.2 in F1@5 and to 3.6, 6.3, and 9.5 in F1@M on three datasets. It shows that One2Branch has better set generation capabilities.

For present keyphrases, they may not be an unordered set in some cases due to the Present-Absent concatenation ordering, which potentially reduces the risk of One2Seq suffering from order bias. For example, the cases whose present keyphrase number is only one are ordered for the present part, and their proportions in the three datasets are 16.7%, 15.6% and 38.1%. Despite this, One2Branch still performs better overall, outperforming One2Seq at least 3.5 in F1@5, and at least 1.7 in F1@M

---

[1] https://github.com/huggingface/transformers

Table 2: Comparison with the sequential decoder of the One2Seq scheme. The reported results are the average scores of three seeds. The standard deviation of each F1 score is presented in the subscript. For example, $33.6_1$ means an average of 33.6 with a standard deviation of 0.1. We reported the throughput and GPU Memory in inference stage with batch size 1.

| | Backbone | Present | | Absent | | Throughput (example/s) | GPU Memory |
|---|---|---|---|---|---|---|---|
| | | F1@5 | F1@M | F1@5 | F1@M | | |
| **KP20k** | | | | | | | |
| One2Seq | T5-Base | $33.6_1$ | $\mathbf{38.8_0}$ | $1.7_0$ | $3.4_0$ | 2.5 | 1,714 M |
| One2Branch | | $\mathbf{36.3_1}$ | $35.2_3$ | $\mathbf{4.7_1}$ | $\mathbf{5.6_1}$ | 6.8 (2.7x) | 1,858 M |
| One2Seq | T5-Large | $34.3_2$ | $\mathbf{39.3_0}$ | $1.7_0$ | $3.5_0$ | 1.6 | 3,972 M |
| One2Branch | | $\mathbf{36.7_1}$ | $38.4_0$ | $\mathbf{5.2_0}$ | $\mathbf{7.1_0}$ | 5.1 (3.2x) | 3,632 M |
| **KPTimes** | | | | | | | |
| One2Seq | T5-Base | $34.6_2$ | $49.2_2$ | $15.3_1$ | $24.2_1$ | 3.0 | 1,534 M |
| One2Branch | | $\mathbf{38.2_2}$ | $\mathbf{51.1_1}$ | $\mathbf{25.7_1}$ | $\mathbf{28.7_5}$ | 7.5 (2.5x) | 2,084 M |
| One2Seq | T5-Large | $36.6_0$ | $50.8_1$ | $15.7_1$ | $24.1_1$ | 1.7 | 3,944 M |
| One2Branch | | $\mathbf{40.1_2}$ | $\mathbf{52.5_2}$ | $\mathbf{27.3_8}$ | $\mathbf{30.4_5}$ | 4.4 (2.6x) | 3,682 M |
| **StackEx** | | | | | | | |
| One2Seq | T5-Base | $28.7_1$ | $\mathbf{56.1_1}$ | $9.4_0$ | $21.6_1$ | 6.6 | 1,476 M |
| One2Branch | | $\mathbf{29.9_3}$ | $55.0_1$ | $\mathbf{19.6_7}$ | $\mathbf{31.1_6}$ | 10.6 (1.6x) | 1,500 M |
| One2Seq | T5-Large | $\mathbf{30.5_2}$ | $\mathbf{58.0_3}$ | $10.6_1$ | $23.9_2$ | 3.7 | 3,486 M |
| One2Branch | | $30.3_1$ | $56.0_1$ | $\mathbf{20.5_2}$ | $\mathbf{32.0_1}$ | 6.3 (1.7x) | 3,472 M |

on KPTimes. It is worth noting that KPTimes has the largest proportion of absent keyphrase compared to the other two datasets (shown in Table 1), which may bring greater challenges to the training of One2Seq with unordered set, resulting in overall performance lagging behind One2Branch. One2Branch performs comparably to One2Seq on KP20k and StackEx with better F1@5 but lower F1@M. On these two datasets, One2Branch generally performs better when the target set is larger.

**Throughput.** One2Branch has significant advantages over One2Seq in inference speed. Benefiting from the parallel decoding capability of the branching decoder, the advantages of One2Branch are more obvious when the target set is larger. Recall the statistics in Table 1, KP20k has the highest average number of keyphrases, so it witnesses the largest speedup accordingly (3.2 times faster than One2Seq based on T5-Large). StackEx has the lowest average number of keyphrases, so the speedup on this dataset is lower than the other two datasets, but still at least 1.6 times faster than One2Seq. Impressively, One2Branch based on T5-Large may even be faster than One2Seq based on T5-Base (5.1 vs. 2.5 on KP20k and 4.4 vs. 3.0 on KPTimes), even though the former has 2–3 times as many parameters as the latter.

**GPU memory usage.** Although the higher throughput of One2Branch comes from generating multiple paths in parallel at the same time, their GPU memory usage is comparable, and One2Branch even uses less memory on the large version. There are two factors that cause this phenomenon. Firstly, the common prefix of multiple sequences is only generated once, while One2Seq needs to generate each one independently. Secondly, the sequence generated by One2Seq needs to interact with the previously generated sequences, which will increase the usage of GPU memory.

## 5.2 Comparison with SOTA models

Following the work of Wu et al., 2022, we compared with two categories of state-of-the-art keyphrase generation models. The first category is to study better model structures and mechanisms to improve sequential decoder. **CatSeq** (Yuan et al., 2020) integrates a copy mechanism (Meng et al., 2017) into the model, and **ExHiRD-h** (Chen et al., 2021) further improves it by using an exclusion mechanism to reduce duplicates. Ye et al. (2021) first introduces **Transformer** (Vaswani et al., 2017) to One2Seq model and then proposes **SetTrans** with an order-agnostic training algorithm for sequential decoder. The second category is to use powerful pre-trained models such as **BART** (Lewis et al., 2020) and **T5** (Raffel et al., 2020), and further pre-train on in-domain data, such as **KeyBart** Kulkarni et al. (2022), **NewsBart** Wu et al. (2022), **SciBart** Wu et al. (2022). The comparison results are reported in Table 3. For the absent keyphrases, One2Branch exceeds all

Table 3: Comparison with state-of-the-art One2Seq models.

| | KP20k | | KPTimes | | StackEx | |
|---|---|---|---|---|---|---|
| | F1@5 | F1@M | F1@5 | F1@M | F1@5 | F1@M |
| **Present** | | | | | | |
| CatSeq | 29.1 | 36.7 | 29.5 | 45.3 | - | - |
| ExHiRD-h | $31.1_1$ | $37.4_0$ | $32.1_{16}$ | $45.2_7$ | $28.8_2$ | $54.8_2$ |
| Transformer | $33.3_1$ | $37.6_2$ | $30.2_5$ | $45.3_6$ | $30.8_5$ | $55.4_2$ |
| SetTrans | $35.6_0$ | $39.1_2$ | $35.6_5$ | $46.3_4$ | $\mathbf{35.8}_3$ | $56.7_5$ |
| BART-Base | $32.2_2$ | $38.8_3$ | $35.9_1$ | $49.9_2$ | $30.4_1$ | $57.1_1$ |
| BART-Large | $33.2_4$ | $39.2_2$ | $37.3_{16}$ | $51.0_{15}$ | $31.2_2$ | $57.8_8$ |
| T5-Base | $33.6_1$ | $38.8_0$ | $34.6_2$ | $49.2_2$ | $28.7_1$ | $56.1_1$ |
| T5-Large | $34.3_2$ | $39.3_0$ | $36.6_0$ | $50.8_1$ | $30.5_2$ | $58.0_3$ |
| KeyBART | $32.5_1$ | $39.8_2$ | $37.8_6$ | $51.3_1$ | $31.9_5$ | $\mathbf{58.9}_2$ |
| SciBART-Base | $34.1_1$ | $39.6_2$ | $34.8_4$ | $48.8_1$ | $30.4_6$ | $57.6_4$ |
| SciBART-Large | $34.7_3$ | $\mathbf{41.5}_4$ | $35.3_4$ | $49.7_2$ | $30.9_3$ | $57.8_2$ |
| NewsBART-Base | $32.4_3$ | $38.7_2$ | $35.4_2$ | $49.8_1$ | $30.7_3$ | $57.5_0$ |
| One2Branch (T5-Base) | $36.3_1$ | $35.2_3$ | $38.2_2$ | $51.1_1$ | $29.9_3$ | $55.0_1$ |
| One2Branch (T5-Large) | $\mathbf{36.7}_1$ | $38.4_0$ | $\mathbf{40.1}_2$ | $\mathbf{52.5}_2$ | $30.3_1$ | $56.0_1$ |
| **Absent** | | | | | | |
| CatSeq | 1.5 | 3.2 | 15.7 | 22.7 | - | - |
| ExHiRD-h | $1.6_0$ | $2.5_0$ | $13.4_2$ | $16.5_1$ | $10.1_1$ | $15.5_1$ |
| Transformer | $2.2_2$ | $4.6_4$ | $17.1_1$ | $23.1_1$ | $10.4_2$ | $18.7_2$ |
| SetTrans | $3.5_1$ | $5.8_1$ | $19.8_3$ | $21.9_2$ | $13.9_1$ | $20.7_0$ |
| BART-Base | $2.2_1$ | $4.2_2$ | $17.1_2$ | $24.9_1$ | $11.7_0$ | $24.9_2$ |
| BART-Large | $2.7_2$ | $4.7_2$ | $17.6_{10}$ | $24.4_{19}$ | $12.4_1$ | $26.1_3$ |
| T5-Base | $1.7_0$ | $3.4_0$ | $15.3_1$ | $24.2_1$ | $9.4_0$ | $21.6_1$ |
| T5-Large | $1.7_0$ | $3.5_0$ | $15.7_1$ | $24.1_1$ | $10.6_1$ | $23.9_2$ |
| KeyBART | $2.6_1$ | $4.7_1$ | $18.0_7$ | $25.5_2$ | $13.0_5$ | $27.1_5$ |
| SciBART-Base | $2.9_3$ | $5.2_4$ | $17.2_3$ | $24.6_2$ | $11.1_6$ | $24.2_8$ |
| SciBART-Large | $3.1_2$ | $5.7_3$ | $17.2_3$ | $25.7_2$ | $12.6_1$ | $26.7_1$ |
| NewsBART-Base | $2.2_1$ | $4.4_2$ | $17.6_3$ | $26.1_1$ | $12.1_3$ | $25.7_4$ |
| One2Branch (T5-Base) | $4.7_1$ | $5.6_1$ | $25.7_1$ | $28.7_5$ | $19.6_7$ | $31.1_6$ |
| One2Branch (T5-Large) | $\mathbf{5.2}_0$ | $\mathbf{7.1}_0$ | $\mathbf{27.3}_8$ | $\mathbf{30.4}_5$ | $\mathbf{20.5}_2$ | $\mathbf{32.0}_1$ |

One2Seq baselines on all three datasets, demonstrating its superior set generation capabilities. For the present keyphrases, One2Branch is also among the best on two datasets.

## 5.3 Additional Experiments

We conducted supplementary experiments, detailed in the appendix. We analysed the effectiveness of training with negative sequences in Section A.1 and the influence of the hyperparameter $k^{\min}$ in Section A.2. Case study is presented in Section A.3 to analyze the characteristics of One2Branch concretely. To more comprehensively assess One2Branch's performance, we experimented with four out-of-distribution datasets (Section A.4) and a question answering dataset (Section A.5). These experiments reinforce the advantages of One2Branch as discussed in Section 5.1.

## 6 Conclusion

In this paper, we propose a new decoder called branching decoder, which can generate a set of sequences in parallel, in contrast with existing decoders that successively generate all the sequences in a single concatenated long sequence. In the experiments, the branching decoder performed impressively in both set generation performance and inference speed.

As a new paradigm for decoders, although we have demonstrated its promising effectiveness and efficiency in this work, more in-depth exploration is still needed. Currently, the branching decoder is implemented based on the existing pre-trained model with inconsistent training methods, which may limit the capability of branching decoder. It would be helpful to explore pre-training methods that better fit the branching decoder. In addition, whether branching decoder can achieve consistent performance on different generation architectures, such as GPT-style causal language model, deserves further experimental analysis.

ACKNOWLEDGMENTS

This work was supported by the NSFC (62072224) and the CAAI-Huawei MindSpore Open Fund.

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

Table 4: The ablation study of negative sequence. Avg denotes the average of F1@5 and F1@M.

| | Backbone | KP20k | | KPTimes | | StackEx | |
|---|---|---|---|---|---|---|---|
| | | F1@5 | F1@M | F1@5 | F1@M | F1@5 | F1@M |
| **Present** | | | | | | | |
| One2Branch | T5-Base | **36.3** | 35.2 | **38.2** | **51.1** | 29.9 | **55.0** |
| w/o negative | | 34.5 | **36.4** | 35.5 | 51.0 | **33.3** | 51.2 |
| One2Branch | T5-Large | **36.7** | **38.4** | **40.1** | **52.5** | 30.3 | **56.0** |
| w/o negative | | 36.5 | 38.2 | 39.2 | 52.4 | **33.4** | 52.3 |
| **Absent** | | | | | | | |
| One2Branch | T5-Base | **4.7** | 5.6 | 25.7 | **28.7** | 19.6 | **31.1** |
| w/o negative | | 4.3 | **5.9** | **26.7** | 27.2 | **22.9** | 24.0 |
| One2Branch | T5-Large | **5.2** | **7.1** | 27.3 | **30.4** | 20.5 | **32.0** |
| w/o negative | | 5.0 | 7.0 | **27.7** | 29.2 | **23.3** | 25.2 |

*New Orleans, Louisiana, USA, February 2-7, 2018*, pp. 5698–5705. AAAI Press, 2018. URL https://www.aaai.org/ocs/index.php/AAAI/AAAI18/paper/view/16784.

Yan Zhang, Jonathon S. Hare, and Adam Prügel-Bennett. Deep set prediction networks. In Hanna M. Wallach, Hugo Larochelle, Alina Beygelzimer, Florence d'Alché-Buc, Emily B. Fox, and Roman Garnett (eds.), *Advances in Neural Information Processing Systems 32: Annual Conference on Neural Information Processing Systems 2019, NeurIPS 2019, December 8-14, 2019, Vancouver, BC, Canada*, pp. 3207–3217, 2019. URL https://proceedings.neurips.cc/paper/2019/hash/6e79ed05baec2754e25b4eac73a332d2-Abstract.html.

Chao Zhao, Marilyn A. Walker, and Snigdha Chaturvedi. Bridging the structural gap between encoding and decoding for data-to-text generation. In Dan Jurafsky, Joyce Chai, Natalie Schluter, and Joel R. Tetreault (eds.), *Proceedings of the 58th Annual Meeting of the Association for Computational Linguistics, ACL 2020, Online, July 5-10, 2020*, pp. 2481–2491. Association for Computational Linguistics, 2020. doi: 10.18653/v1/2020.acl-main.224. URL https://doi.org/10.18653/v1/2020.acl-main.224.

# A APPENDIX

## A.1 ABLATION STUDY: TRAINING WITH NEGATIVE SEQUENCES

A unique feature of One2Branch is its training with negative sequences. In this section, we delve into the effectiveness of this approach. For a fair comparison, we trained the checkpoints from the first stage for an additional 5 epochs without using negative sequences. Table 4 reports the ablation results. The incorporation of negative sequences resulted in performance improvement in most cases (16 out of 24). Among them, the absent keyphases of StackEx showed the most significant increase on F1@M, recording 7.1 for the base version and 6.8 for the large version. Although using negative sequences induces a drop of 8 indicators, the technique holds promise. Exploring it further, for instance, through dynamic negative sequence sampling as suggested by Xiong et al. (2021), might offer enhancements in the future.

## A.2 INFLUENCE OF THE MINIMUM NUMBER OF EXPLORED PATHS

To prevent One2Branch from inadequate generation, we set a minimum for explored paths, denoted as $k^{min}$, as detailed in section 3.4. Instead of requiring a score strictly greater than 0 at every time-step, we allow the average generation score to be above 0. This adjustment, meaning a larger $k^{min}$, allows the branching decoder to explore a broader generation space. Figure 4 shows the average number of keyphrases (# KP) generated by One2Branch; as $k^{min}$ increases, # KP also increases. For StackEx, when $k^{min}$ is greater than 5, # KP changes slowly, indicating that a larger exploration space is not necessary. For KP20k and KPTimes, # KP is still a growing trend when $k^{min} = 15$.

In order to further analyze the effect of $k^{min}$ on generation performance, we presented the F1 score of One2Branch on the test sets in Figure 5 and Figure 6. As $k^{min}$ increases, the F1@5 of both present

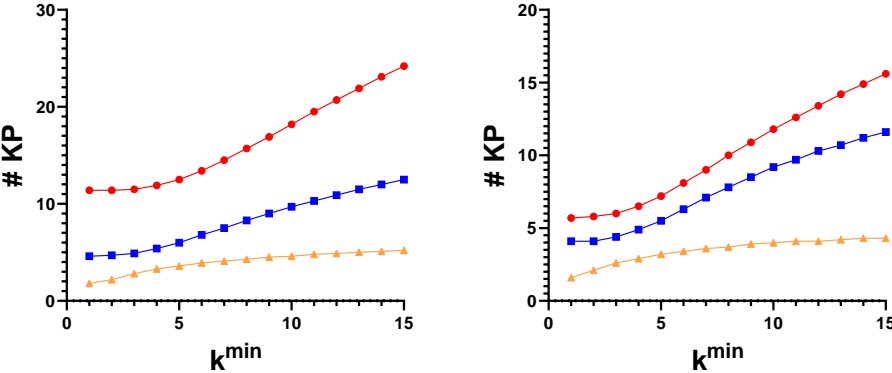

Figure 4: The average number of keyphrases generated by One2Branch-Base (left) and One2Branch-Large (right) under different $k^{\min}$. Red (dot), blue (square), and orange (triangle) refer to the KP20k, KPTimes, and StackEx dataset, respectively.

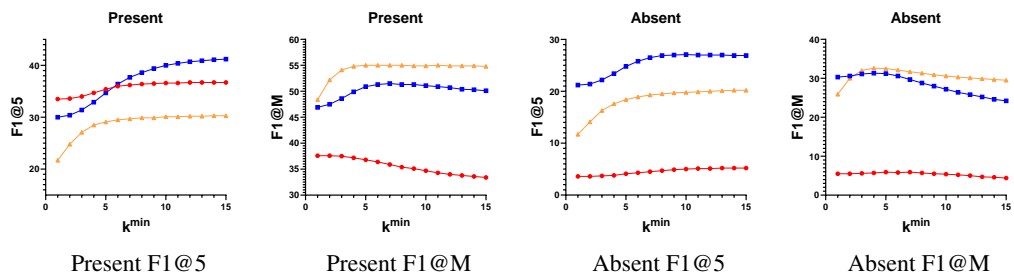

| Present F1@5 | Present F1@M | Absent F1@5 | Absent F1@M |

Figure 5: The keyphrase generation performance of One2Branch (T5-Base) under different $k^{\min}$. Red (dot), blue (square), and orange (triangle) refer to the KP20k, KPTimes, and StackEx dataset, respectively.

and absent keyphrases gradually increases and then flattens, indicating that a larger exploration space helps generate correct sequences for the samples with more target keyphrases. From the performance on KPTimes and StackEx, we note that as $k^{\min}$ increases, F1@M first increases and then decreases. This suggests that moderately increasing $k^{\min}$ can enhance exploration and generation. However, an excessively large $k^{\min}$ can introduce noise.

## A.3 CASE STUDY

Table 5 presents a case from the test set of KP20k generated by One2Branch-Large with $k^{\min}$ set to 1, 8, and 15. Compared with $k^{\min} = 1$, a larger $k^{\min}$ can help the model generate more insightful keyphrases. For example, when $k^{\min} \geq 8$, One2Branch generates the absent keyphrase "cryptography" which only obscurely expresses in the document.

There are many similar expressions in the generated keyphrases. For example, the generated sequences {"message authentication code", "message authentication", "authentication", "mac scheme", "mac"} all have the same meaning and some of them have the same prefix. One2Branch would generate multiple semantically similar sequences when it is difficult to determine the optimal one. This problem may be alleviated by adding a deduplication module. In addition, an excessively large $k^{\min}$ can lead to overly broad generation. For example, when $k^{\min} = 15$, the generated keyphrase "digital signature" is irrelevant to the document. Therefore, a suitable $k^{\min}$ is integral to One2Branch.

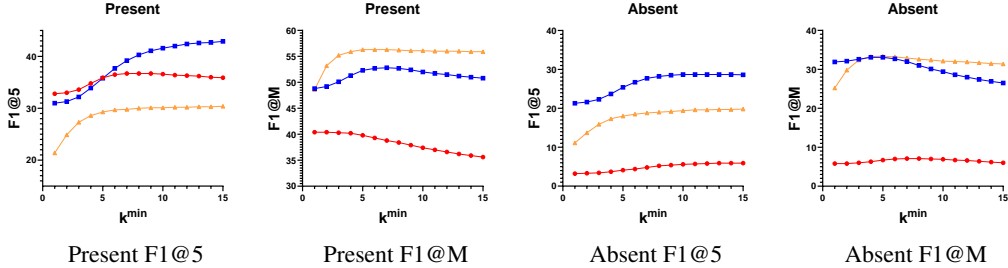

| Present F1@5 | Present F1@M | Absent F1@5 | Absent F1@M |
|:---:|:---:|:---:|:---:|

Figure 6: The keyphrase generation performance of One2Branch (T5-Large) under different $k^{min}$. Red (dot), blue (square), and orange (triangle) refer to the KP20k, KPTimes, and StackEx dataset, respectively.

Table 5: A case generated by One2Branch from KP20k. The keyphrases generated correctly are marked blue.

| | |
|---|---|
| Document | Construct message authentication code with **one way hash functions** and **block ciphers**. We suggest an **mac** scheme which combines a hash function and an block cipher in order. We strengthen this scheme to prevent the problem of leaking the intermediate hash value between the hash function and the block cipher by additional random bits. The requirements to the used hash function are loosely. Security of the proposed scheme is heavily dependent on the underlying block cipher. This scheme is efficient on software implementation for processing long messages and has clear security properties. |
| Ground-truth | **Present**: {one way hash function, block cipher, mac} 
 **Absent**: {cryptography} |
| $k^{min} = 1$ 
 Generated KPs | {block cipher, message authentication code, message authentication, one way hash function} |
| $k^{min} = 8$ 
 Generated KPs | {block cipher, message authentication code, message authentication, one way hash function, hash function, authentication, authentication code, mac scheme, cryptography, security proof, mac} |
| $k^{min} = 15$ 
 Generated KPs | {block cipher, message authentication code, message authentication, one way hash function, hash function, authentication, authentication code, mac scheme, cryptography, security proof, mac, digital signature} |

Table 6: Statistics of keyphrase generation (KG) and question answering (QA) datasets. # Trg, |Trg|, and % Abs Trg refer to the average number of target sequences per document, the average number of words that each target sequence contains, and the percentage of absent target sequences, respectively. All of them are calculated over the dev set.

| Dataset | Task | # Test | # Trg | \|Trg\| | % Abs Trg |
|---|---|---|---|---|---|
| Inspec | KG | 500 | 5.3 | 2.0 | 37.1 |
| Krapivin | KG | 400 | 9.8 | 2.5 | 26.4 |
| NUS | KG | 211 | 5.9 | 2.2 | 44.3 |
| SemEval | KG | 100 | 14.7 | 2.4 | 57.4 |
| MSQA | QA | 653 | 2.9 | 3.0 | 0 |

## A.4 OUT-OF-DISTRIBUTION KEYPHRASE GENERATION DATASETS

Following Wu et al. (2022), we used the One2Branch model trained on KP20k to evaluate on four out-of-distribution datasets. Some statistics of these datasets are shown in Table 6.

Table 7 reports the comparsion of One2Branch and One2Seq in term of generation performance, inference speed, and GPU memory usage. Similar to the results in Section 5.1, One2Branch has outstanding performance in the generation of absent keyphrase and inference speed. It is comparable and alternately leads with One2Seq on present keyphrases.

Table 7: Comparison with the sequential decoder of the One2Seq scheme on out-of-domain keyphrase generation datasets. The reported results are the average scores over three seeds. The standard deviation of each F1 score is presented in the subscript. For example, $33.6_1$ means an average of 33.6 with a standard deviation of 0.1. We reported the throughput and GPU Memory in inference stage with batch size 1.

| | Backbone | Present | | Absent | | Throughput (example/s) | GPU Memory |
|---|---|---|---|---|---|---|---|
| | | F1@5 | F1@M | F1@5 | F1@M | | |
| **Inspec** | | | | | | | |
| One2Seq | T5-Base | $28.8_5$ | $\mathbf{33.9_5}$ | $1.1_1$ | $2.0_3$ | 3.2 | 1,698 M |
| One2Branch | | $\mathbf{30.4_6}$ | $33.7_3$ | $\mathbf{2.6_1}$ | $\mathbf{3.4_2}$ | 8.7 (2.7x) | 1,652 M |
| One2Seq | T5-Large | $29.5_1$ | $\mathbf{34.3_4}$ | $1.1_3$ | $2.1_6$ | 2.1 | 3,948 M |
| One2Branch | | $\mathbf{32.2_1}$ | $34.2_1$ | $\mathbf{3.1_0}$ | $\mathbf{4.0_0}$ | 7.2 (3.4x) | 3,516 M |
| **Krapivin** | | | | | | | |
| One2Seq | T5-Base | $\mathbf{30.2_3}$ | $\mathbf{35.0_2}$ | $2.3_2$ | $4.3_4$ | 2.8 | 1,716 M |
| One2Branch | | $27.5_3$ | $26.9_5$ | $\mathbf{5.0_1}$ | $\mathbf{6.1_3}$ | 9.0 (3.2x) | 1,710 M |
| One2Seq | T5-Large | $\mathbf{31.5_2}$ | $\mathbf{35.9_5}$ | $2.3_4$ | $4.5_7$ | 1.9 | 3,960 M |
| One2Branch | | $30.3_1$ | $31.3_1$ | $\mathbf{5.0_0}$ | $\mathbf{7.9_1}$ | 6.7 (3.5x) | 3,614 M |
| **NUS** | | | | | | | |
| One2Seq | T5-Base | $38.8_6$ | $\mathbf{44.0_4}$ | $2.7_0$ | $5.1_3$ | 3.2 | 1,712 M |
| One2Branch | | $\mathbf{39.1_2}$ | $39.7_4$ | $\mathbf{5.6_7}$ | $\mathbf{6.5_6}$ | 9.3 (2.9x) | 1,706 M |
| One2Seq | T5-Large | $39.8_4$ | $\mathbf{43.8_6}$ | $2.5_3$ | $4.2_6$ | 1.9 | 3,944 M |
| One2Branch | | $\mathbf{41.6_1}$ | $43.6_0$ | $\mathbf{6.0_0}$ | $\mathbf{8.3_0}$ | 6.8 (3.6x) | 3,592 M |
| **SemEval** | | | | | | | |
| One2Seq | T5-Base | $\mathbf{29.5_{16}}$ | $\mathbf{32.6_{16}}$ | $1.4_4$ | $2.0_5$ | 3.6 | 1,712 M |
| One2Branch | | $29.3_2$ | $30.6_4$ | $\mathbf{2.5_1}$ | $\mathbf{2.9_5}$ | 8.7 (2.4x) | 1,618 M |
| One2Seq | T5-Large | $\mathbf{29.7_{10}}$ | $\mathbf{32.1_{11}}$ | $1.5_1$ | $2.0_3$ | 1.9 | 3,974 M |
| One2Branch | | $28.2_0$ | $31.5_1$ | $\mathbf{2.7_0}$ | $\mathbf{3.4_1}$ | 7.1 (3.7x) | 3,560 M |

In Table 8, we reported the comparsion of One2Branch with state-of-the-art One2Seq models. It can be seen that One2Branch achieves a significant lead in absent keyphrase generation compared to all baselines. For present keyphrase, SetTrans is still the best performing model, thanks to its ability of capturing the dependencies between generated sequences.

## A.5  QA PERFORMANCE

Besides the keyphrase generation datasets, we also experimented with One2Branch on a multi-answer question answering dataset MSQA (Li et al., 2022). Each sample in MSQA includes a question with at least two answers, and a document containing all answers is given. Some dataset statistics are reported in Table 6.

For the implementation of both One2Branch and One2Seq, we searched the best learning rate from {1e-4, 3e-4} and trained models 50 epochs. The maximum sequence length was set to 2048 to ensure that all documents are not truncated. For One2Seq, we concatenated the answers in the order of their occurrence in the document. We followed Li et al. (2022) to use two metrics. Exact Match (**EM**) measures the F1 score between generated answers and gold-standard answers, which requires an exact match between answer texts. Partial Match (**PM**) generalizes EM by using the length of the longest common substring to score each predicted answer based on its nearest ground-truth answer.

We experimented with MSQA in two configurations, with and without the given document. When using the document, MSQA becomes an extraction task, extracting all answers from the document. It is worth noting that the dataset provides the order in which the answers appear in the document. This is no longer an unordered set generation task, which allows the model to generate answers in order. When the document is not used, MSQA becomes a fully absent generation task, so that models need to generate answers from its parameters.

The comparison results of One2Branch and One2Seq on MSQA are reported in Table 9. One2Branch lags slightly behind One2Seq in the configuration of using document, with a maximum

Table 8: Comparison with state-of-the-art One2Seq models on four out-of-distribution datasets. All of the models were trained on the KP20k dataset.

| | Inspec | | Krapivin | | NUS | | SemEval | |
|---|---|---|---|---|---|---|---|---|
| | F1@5 | F1@M | F1@5 | F1@M | F1@5 | F1@M | F1@5 | F1@M |
| **Present** | | | | | | | | |
| CatSeq | 22.5 | 26.2 | 26.9 | 35.4 | 32.3 | 39.7 | 24.2 | 28.3 |
| ExHiRD-h | $25.4_4$ | $29.1_3$ | $28.6_4$ | $30.8_4$ | - | - | $30.4_{17}$ | $28.2_{18}$ |
| Transformer | $28.8_7$ | $33.3_5$ | $31.4_9$ | $36.5_7$ | $37.8_6$ | $42.9_9$ | $28.8_5$ | $32.1_8$ |
| SetTrans | $29.1_3$ | $32.8_1$ | $\mathbf{33.5}_{10}$ | $\mathbf{37.5}_{11}$ | $39.9_8$ | $\mathbf{44.6}_{22}$ | $\mathbf{32.2}_8$ | $\mathbf{34.2}_{14}$ |
| BART-Base | $27.0_3$ | $32.3_7$ | $27.0_6$ | $33.6_6$ | $36.6_1$ | $42.4_8$ | $27.1_{11}$ | $32.1_{21}$ |
| BART-Large | $27.6_{11}$ | $33.3_9$ | $28.4_2$ | $34.7_3$ | $38.0_8$ | $43.5_{11}$ | $27.4_{12}$ | $31.1_{16}$ |
| T5-Base | $28.8_5$ | $33.9_5$ | $30.2_3$ | $35.0_2$ | $38.8_6$ | $44.0_4$ | $29.5_{16}$ | $32.6_{16}$ |
| T5-Large | $29.5_1$ | $\mathbf{34.3}_4$ | $31.5_2$ | $35.9_5$ | $39.8_4$ | $43.8_6$ | $29.7_{10}$ | $32.1_{11}$ |
| KeyBART | $26.8_3$ | $32.5_5$ | $28.7_6$ | $36.5_{14}$ | $37.3_7$ | $43.0_{10}$ | $26.0_8$ | $28.9_4$ |
| SciBART-Base | $27.5_{10}$ | $32.8_8$ | $28.2_8$ | $32.9_{11}$ | $37.3_7$ | $42.1_{14}$ | $27.0_8$ | $30.4_8$ |
| SciBART-Large | $26.1_{12}$ | $31.7_{13}$ | $27.1_{11}$ | $32.4_{12}$ | $36.4_{18}$ | $40.9_{12}$ | $27.9_{14}$ | $32.0_{12}$ |
| NewsBART-Base | $26.2_{10}$ | $31.7_{11}$ | $26.2_8$ | $32.3_{15}$ | $36.9_8$ | $42.4_{10}$ | $26.4_{21}$ | $30.4_{23}$ |
| One2Branch (T5-Base) | $30.4_6$ | $33.7_3$ | $27.5_3$ | $26.9_5$ | $39.1_2$ | $39.7_4$ | $29.3_2$ | $30.6_4$ |
| One2Branch (T5-Large) | $\mathbf{32.2}_1$ | $34.2_1$ | $30.3_1$ | $31.3_1$ | $\mathbf{41.6}_1$ | $43.6_0$ | $28.2_0$ | $31.5_1$ |
| **Absent** | | | | | | | | |
| CatSeq | 0.4 | 0.8 | 1.8 | 3.6 | 1.6 | 2.8 | 2.0 | 2.8 |
| ExHiRD-h | $1.1_1$ | $1.6_2$ | $2.2_3$ | $3.3_4$ | - | - | $1.6_4$ | $2.1_6$ |
| Transformer | $1.2_0$ | $2.3_1$ | $3.3_2$ | $6.3_4$ | $2.5_4$ | $4.4_9$ | $1.6_2$ | $2.2_4$ |
| SetTrans | $1.9_1$ | $3.0_1$ | $4.5_1$ | $7.2_3$ | $3.7_{10}$ | $5.5_{17}$ | $2.2_2$ | $2.9_2$ |
| BART-Base | $1.0_1$ | $1.7_2$ | $2.8_3$ | $4.9_6$ | $2.6_4$ | $4.2_9$ | $1.6_1$ | $2.1_2$ |
| BART-Large | $1.5_3$ | $2.4_4$ | $3.1_1$ | $5.1_2$ | $3.1_5$ | $4.8_9$ | $1.9_3$ | $2.4_3$ |
| T5-Base | $1.1_1$ | $2.0_3$ | $2.3_2$ | $4.3_4$ | $2.7_0$ | $5.1_3$ | $1.4_4$ | $2.0_5$ |
| T5-Large | $1.1_3$ | $2.1_6$ | $2.3_4$ | $4.5_7$ | $2.5_3$ | $4.2_6$ | $1.5_1$ | $2.0_3$ |
| KeyBART | $1.4_2$ | $2.3_2$ | $3.6_2$ | $6.4_6$ | $3.1_4$ | $5.5_7$ | $1.6_4$ | $2.2_5$ |
| SciBART-Base | $1.6_2$ | $2.8_4$ | $3.3_4$ | $5.4_8$ | $3.3_1$ | $5.3_2$ | $1.8_1$ | $2.2_1$ |
| SciBART-Large | $1.5_2$ | $2.6_2$ | $3.4_1$ | $5.6_3$ | $3.2_5$ | $5.0_7$ | $2.6_6$ | $3.3_8$ |
| NewsBART-Base | $1.0_1$ | $1.8_2$ | $2.4_2$ | $4.5_4$ | $2.4_4$ | $4.0_9$ | $1.6_1$ | $2.2_2$ |
| One2Branch (T5-Base) | $2.6_1$ | $3.4_2$ | $\mathbf{5.0}_1$ | $6.1_3$ | $5.6_7$ | $6.5_6$ | $2.5_1$ | $2.9_5$ |
| One2Branch (T5-Large) | $\mathbf{3.1}_0$ | $\mathbf{4.0}_0$ | $\mathbf{5.0}_0$ | $7.9_1$ | $\mathbf{6.0}_0$ | $\mathbf{8.3}_0$ | $\mathbf{2.7}_0$ | $3.4_1$ |

Table 9: Comparison with the sequential decoder of the One2Seq scheme on the multi-span question answering dataset (MSQA). The reported results are the average scores over three seeds. The standard deviation of each F1 score is presented in the subscript. For example, $72.3_4$ means an average of 72.3 with a standard deviation of 0.4. We reported the throughput and GPU Memory in inference stage with batch size 1.

| | Backbone | Dev | | Test | | Throughput | GPU |
|---|---|---|---|---|---|---|---|
| | | EM | PM | EM | PM | (example/s) | Memory |
| **w/ doc (extraction task)** | | | | | | | |
| One2Seq | T5-Base | $\mathbf{72.3}_4$ | $\mathbf{84.0}_1$ | $\mathbf{71.5}_8$ | $\mathbf{84.1}_6$ | 5.3 | 2,220 M |
| One2Branch | | $71.4_1$ | $83.2_2$ | $70.2_3$ | $82.9_5$ | 9.0 (1.7x) | 2,060 M |
| One2Seq | T5-Large | $74.6_2$ | $\mathbf{85.9}_4$ | $\mathbf{74.7}_7$ | $\mathbf{86.4}_3$ | 3.2 | 4,986 M |
| One2Branch | | $\mathbf{74.7}_3$ | $85.7_3$ | $73.3_6$ | $85.5_6$ | 5.7 (1.8x) | 4,640 M |
| **w/o doc (generation task)** | | | | | | | |
| One2Seq | T5-Base | $16.4_4$ | $35.6_2$ | $16.0_4$ | $35.4_5$ | 7.2 | 2,346 M |
| One2Branch | | $\mathbf{19.9}_2$ | $\mathbf{36.5}_5$ | $\mathbf{20.9}_4$ | $\mathbf{37.2}_2$ | 8.9 (1.2x) | 1,480 M |
| One2Seq | T5-Large | $18.2_2$ | $37.3_4$ | $18.1_2$ | $37.3_1$ | 4.8 | 3,878 M |
| One2Branch | | $\mathbf{21.6}_2$ | $\mathbf{38.8}_3$ | $\mathbf{23.3}_5$ | $\mathbf{39.7}_5$ | 8.0 (1.7x) | 3,352 M |

gap of 1.4 on the EM of the test set. This may be a benefit of One2Seq in which later generated sequences can interact with previously generated sequences. Moreover, this is not a set generation task because the order of the answers is known. However, One2Branch has obvious advantages in inference speed. The inference time of One2Branch based on T5-Large is even better than One2Seq based on T5-Base, when the former has significant advantage in QA performance.

One2Branch achieves more significant lead in the configuration of without using document, with a maximum gap of 5.2 on the EM of the test set. This demonstrates the potential of One2Branch a broader range of set generation tasks beyond keyphrase generation.

