# OpenReview forum: "A Branching Decoder for Set Generation"
_ICLR.cc/2024/Conference — ICLR 2024 poster_

### Official Review · Reviewer_4BdG · 2023-10-29

**Soundness:** 4 excellent
**Presentation:** 3 good
**Contribution:** 3 good
**Rating:** 6
**Confidence:** 4

**Summary:**

The paper considers set generation with seq2seq models. The proposed solution dubbed One2Branch is a combination of parallel decoding and the ZLPR loss which allows for a dynamic number of hypotheses. For training, a stepwise version of the ZLPR loss is optimized for both positive and negative label sequences. For inference, path-level scores are used with min/max number of paths. One2Branch is shown to outperform direct linearization (One2Seq) on keyphrase generation.

**Strengths:**

- Substantially novel approach to seq2seq set generation
- Well-performing

**Weaknesses:**

- The method yields a task-specific model specialized for set generation (i.e., it loses the generality of seq2seq, in contrast to One2Seq).
- The improvement is not very large, since the baselines are already pretty good.

**Questions:**

While parallel decoding is more efficient, does it not also lose the benefit of autoregressive reasoning as well? It seems reasonable to expect that a sufficiently powerful One2Seq model may ultimately perform better than a more specialized set generation model.

---

> ### Author Response · Authors · 2023-11-17
>
> Many thanks for acknowledging the novelty and soundness of our work.
>
> The reviewer might have some misunderstanding of the generalizability of our approach, which we will clarify first in the following.
>
> *W1: "The method yields a task-specific model specialized for set generation (i.e., it loses the generality of seq2seq, in contrast to One2Seq)."*
>
> We would like to clarify that our One2Branch **is not specific to set generation but, on the contrary, is a generalization of Seq2Seq**. It has the capability to generate a set of sequences. If needed, it can also be easily configured to work as a Seq2Seq model by either only taking the highest-scored sequence from its output set, or fixing $k^\text{min}=k^\text{max}=1$ in Algorithm 1. We will incorporate this clarification into the camera-ready version.
>
> *W2: "The improvement is not very large, since the baselines are already pretty good."*
>
> Please note that our improvement is twofold: in accuracy and in throughput. **Both improvements are considerable.**
>
> For accuracy, the baselines are actually not that good. For example, as shown in Table 2, for absent keyphrases, our One2Branch improves over the baseline One2Seq **by up to 3.5, 11.6, and 10.2 in F1@5** on three datasets, representing fairly large improvements in accuracy.
>
> For throughput, our One2Branch is **3.2 times faster** than the baseline One2Seq on the KP20k dataset, and is **1.6 times faster** than One2Seq on the StackEx dataset, representing a considerable improvement in throughput.
>
> *Q1: "While parallel decoding is more efficient, does it not also lose the benefit of autoregressive reasoning as well? It seems reasonable to expect that a sufficiently powerful One2Seq model may ultimately perform better than a more specialized set generation model."*
>
> Thanks for this inspiring question.
>
> Our parallel decoding **does not lose the benefit of autoregressive reasoning**. Indeed, recall that our decoder in Equation (2) is still an autoregressive decoder, so the generation of each individual sequence in our model still follows an autoregressive manner and benefits its reasoning capability. Therefore, in theory, our model will be equally effective if used as a Seq2Seq model (see our response to W1).
>
> What our model **intentionally ignores** is the dependence among the generation of different sequences in the set; multiple sequences are independently generated in parallel. For set generation tasks, such decoupling is considered beneficial; it is exactly the motivation of our work. The benefit of decoupling is demonstrated by our experimental results in Table 2 where even the Base version of our model outperforms the Large version of One2Seq in most settings, both in accuracy and in throughput. Therefore, we believe that our One2Branch is a promising generalization of One2Seq.

---

### Official Review · Reviewer_wNPk · 2023-10-30

**Soundness:** 3 good
**Presentation:** 3 good
**Contribution:** 3 good
**Rating:** 6
**Confidence:** 3

**Summary:**

This paper proposes a branching decoder, which can generate a dynamic number of tokens at each time-step and branch multiple generation paths. In particular, paths are generated individually so that no order dependence is required. Moreover, multiple paths can be
generated in parallel which greatly reduces the inference time. The experiments are promising

**Strengths:**

*  The overall idea and motivation are novel for set generation tasks.
* Thorough experiments demonstrate clear benefits over established approaches.
* The training method and inference algorithm are well-designed.
* Strong results on multiple datasets make a compelling case for the branching decoder.

**Weaknesses:**

* Little ablation to analyze the impact of different design choices.
* Limited analysis of how performance varies across different set sizes and domains.
* The factorization of decoder embedding seems unnecessary given modern hardware.

**Questions:**

NA

---

> ### Author Response · Authors · 2023-11-17
>
> Many thanks for the positive comments on novelty, soundness, and evaluation. Below we clarify about the weaknesses mentioned in the review.
>
> *W1 "Little ablation to analyze the impact of different design choices."*
>
> Our approach has two major components: training strategies (Section 3.3) and decoding algorithm (Section 3.4). They are dependent on each other and cannot be ablated. The only ablatable module in our approach is the generation and use of negative sequences for training. **We have already presented an ablation study** of this module in Appendix A.1. Another variable part of our approach is the hyperparameter $k^\text{min}$ in the decoding algorithm. **We have analyzed its influence** in Appendix A.2.
>
> We would appreciate if the reviewer could provide concrete suggestions about which of the remaining components of our approach can or should be ablated for analysis.
>
> *W2 "Limited analysis of how performance varies across different set sizes and domains."*
>
> As to domains, please note that **we have already used four datasets from different domains** in the experiments: science (KP20k), news (KPTimes), forum (StackEx), and open-domain (MSQA), and we drew consistent conclusions.
>
> As to set size, we empirically observed that the performance of our approach generally increases when the size of the gold-standard set increases, possibly because large sets are often associated with relatively easy samples. **This performance result with a varying set size is in ready availability. Following the reviewer's suggestion, we will easily add it to the appendix of the camera-ready version.**
>
> *W3 "The factorization of decoder embedding seems unnecessary given modern hardware."*
>
> We did not perform embedding/matrix factorization in our decoder. Maybe we did not fully understand the comment. We would appreciate if the reviewer could clarify the question.
>
> A possibly related fact is that considering the slow speed of long sequence generation even on modern hardware, our proposed branching decoder generates text in parallel which accelerates inference by several times without additional GPU requirements, which we believe is beneficial to user experience.

---

### Official Review · Reviewer_KcLG · 2023-10-30

**Soundness:** 3 good
**Presentation:** 4 excellent
**Contribution:** 3 good
**Rating:** 8
**Confidence:** 4

**Summary:**

This paper tackles the challenge of set generation which has been mainly formulated as a One2Seq problem. Bypassing the order bias induced in the latter, the authors propose instead a branching decoder that generates sequences in parallel that meet certain requirements. Experiments on keyphrase generation show the effectiveness of their method compared to sequential decoders.

**Strengths:**

- The paper is well-written and the presentation is clear.
- Experiments on keyphrase generation demonstrate strong performance on all datasets compared to SOTA as well as better OOD capabilities.
- Additional experiments show promises for other generation tasks e.g. MSQA.
- Ablation study is conducted on the impact of negative sequences during training.

**Weaknesses:**

see questions

**Questions:**

- I am not familiar with the multi-decoder technique, can you elaborate on what "a decoder with shared parameters" means or provide a reference ? In the text, you mention that "each sequence is input to a decoder independently" does that mean there are multiple decoders with shared parameters ? or it is the exact same decoder that the inputs are fed into, although independently ?
- What loss is the model trained on ? Is it only optimized on the ZPLR loss ? If so, what happens if only positive labels are considered ? How does your method compare with simply selecting the highest scoring sequences instead of selecting sequences that have positive scores in average ? During decoding, the $k_{min}$ highest-scored tokens are guaranteed to be selected at each iteration anyway. Does your proposed method work without the ZPLR loss ?
- Is your negative sequence a contribution ? It would be insightful to have a pointer towards some negative contrastive learning literature that you drew inspiration from and indicate the differences.
- What value of $k_{min}$ is chosen for the computation of throughput / GPU memory usage ? How do they vary with different values of $k_{min}$ ?

**Typo**
- There seems to be a typo in the figure 2: dotted arrow should go from B to CB.

---

> ### Author Response · Authors · 2023-11-17
>
> Many thanks for the time of the reviewer. The questions (mainly confusions) raised in the review can be easily clarified. We will incorporate the following clarifications into the camera-ready version.
>
> *Q1 "I am not familiar with the multi-decoder technique, can you elaborate on what "a decoder with shared parameters" means or provide a reference ? In the text, you mention that "each sequence is input to a decoder independently" does that mean there are multiple decoders with shared parameters ? or it is the exact same decoder that the inputs are fed into, although independently ?"*
>
> We clarify that it is the exact same decoder that the inputs are independently fed into.
>
> *Q2.1 "What loss is the model trained on ? Is it only optimized on the ZPLR loss ? If so, what happens if only positive labels are considered ?"*
>
> Our model is only optimized on the ZPLR loss. It cannot be trained without negative labels. Please note the **difference between negative label (i.e., token) and negative sequence**. Both positive and negative tokens can be straightforwardly derived from positive sequences; they complement to each other and they collectively form the entire vocabulary. They are both needed for training. However, our model can be trained without negative sequences. Indeed, as mentioned at the end of Section 3.3, we firstly train our model only using positive sequences (i.e., without using negative sequences, but only using positive and negative tokens derived from positive sequences).
>
> *Q2.2 "How does your method compare with simply selecting the highest scoring sequences instead of selecting sequences that have positive scores in average ? Does your proposed method work without the ZPLR loss ?"*
>
> The difference is that it would be challenging to determine **how many** highest scoring sequences should be selected. Our model design allows to **dynamically** choose this number (by comparing each score with zero), which represents a distinguishing advantage of our approach. **This advantage is sourced from the ZPLR loss**, which therefore is a key component of our approach.
>
> *Q3 "Is your negative sequence a contribution ? It would be insightful to have a pointer towards some negative contrastive learning literature that you drew inspiration from and indicate the differences."*
>
> Negative sequence is part of our contribution to training our proposed branching decoder.
>
> **Following the reviewer's suggestion, we will add the following discussion to the camera-ready version.** Recall that a common way to train a decoder is to calculate the loss of positive sequences using teacher forcing, which often leads to exposure bias and affects model generalizability [1]. To reduce this bias, contrastive learning calculates a sequence-level loss to supervise the model to distinguish between positive and negative sequence representations [2][3]. By contrast, our branching decoder calculates the loss at the token level, which is believed to be more consistent with autoregressive generation than sequence-level losses.
>
> [1] Scheduled sampling for sequence prediction with recurrent neural networks, NeurIPS 2015
>
> [2] Contrastive Learning with Adversarial Perturbations for Conditional Text Generation, ICLR 2021
>
> [3] CONT: Contrastive Neural Text Generation, NeurIPS 2022
>
> *Q4. "What value of $k^\text{min}$ is chosen for the computation of throughput / GPU memory usage? How do they vary with different values of $k^\text{min}$?"*
>
> As mentioned in Section 4.3, we tuned $k^\text{min} \in [1,15]$ on the dev set and finally fixed $k^\text{min}=8$.
>
> **A smaller value of $k^\text{min}$ would allow a smaller number of paths to be explored, thus increasing throughput.** For example, based on T5-Base on the KP20k dataset, the throughput of our One2Branch model is 7.6 at $k^\text{min}=1$, being 1.1 times as fast as the throughput of 6.8 at $k^\text{min}=8$. The throughput further decreases to 5.6 at $k^\text{min}=15$, but still, it is much faster than the baseline One2Seq. **The influence of varying $k^\text{min}$ on GPU memory usage is negligible.** We will add these results to Appendix A.2 where we have reported the influence of $k^\text{min}$ on F1 scores.

---

### Official Review · Reviewer_JQ4Q · 2023-11-06

**Soundness:** 3 good
**Presentation:** 4 excellent
**Contribution:** 3 good
**Rating:** 8
**Confidence:** 4

**Summary:**

Often generative models use a sequential decoder that generates a single output sequence. For the tasks where multiple outputs are possible (for example, a set generation problem), a popular way is to train a sequential decoder concatenating all outputs in a long sequence. This setup suffers from several limitations.

The paper introduces a branching decoder that can generate branch-out multiple generation paths. One very interesting contribution of the paper is the integration of the ZLPR loss (Su et al., 2022) that allows a threshold-based decoding algorithm (instead of heuristic approaches) for inference. During decoding the branching decoder generates new generative branches identifying  a dynamic set of tokens with logits exceeding a threshold (0 in this case).

The experiments are mainly done on keyphrase generation tasks focusing on three different datasets focusing on different domains and keyphrase lengths (avg 1.3 to 2.2 words on average). The results demonstrate that the branching decoder performs considerably better than the sequential decoders.

--

Thank you for your response. Thanks for including Diverse beam search comparisons. I have updated my reviews accordingly.

**Strengths:**

The novel branching decoder that allows generating multiple target sequences in parallel, generating a dynamic number of tokens at each time-step.

The integration of the ZLPR loss during training allows 1) training with both target and negative sequence, and 2) a threshold-based decoding algorithm (instead of heuristic approaches) for inference.

Solid results showing the advantages of the branch decoders for keyphrase generation tasks.

The codes are made available.

**Weaknesses:**

I believe that there are two major weaknesses in the paper. Addressing them would improve the impact of the paper significantly.

Firstly, the sota comparisons are focused on pretrained models and sequential decoders. Diverse decoding and generation methods (see some relevant papers below)  would also potentially be good for set generation tasks, and it would be interesting to know how branching decoders perform against them.

The other major weakness of the work is that the authors solely focus on the keyphrase generation problem, where the keyphrases are 1-3 words long which is very small compare to natural generation tasks. Natural language generation is inherently a set generation problem, for example, there could be multiple paraphrases for a sentence (paraphrase generation), same questions can be asked differently (question generation) and summary can be written differently (summarization). It would be very interesting to see how branch decoders perform on some of these tasks (see some relevant papers below).

Relevant papers:

https://doi.org/10.18653/v1/D19-1308

http://arxiv.org/abs/1703.06029

https://www.aclweb.org/anthology/2020.findings-emnlp.218

https://arxiv.org/pdf/2105.11921.pdf

http://arxiv.org/abs/1611.08562

https://www.aaai.org/ocs/index.php/AAAI/AAAI18/paper/view/17329

**Questions:**

It would be great to hear the authors’ response to two main weaknesses raised above.

Minor: Also, in section 3.4, why do we need the minimum number of explored paths (k^min) given we are using the threshold-based decoding algorithm?

---

> ### Author Response · Authors · 2023-11-20
>
> Many thanks for the constructive comments.
>
> We would like to firstly address a potential misunderstanding in the review. Then we answer the remaining questions.
>
> *W2. "The other major weakness of the work is that the authors solely focus on the keyphrase generation problem, where the keyphrases are 1-3 words long which is very small compare to natural generation tasks. Natural language generation is inherently a set generation problem, for example, there could be multiple paraphrases for a sentence (paraphrase generation), same questions can be asked differently (question generation) and summary can be written differently (summarization). It would be very interesting to see how branch decoders perform on some of these tasks (see some relevant papers below)."*
>
> We would like to clarify that in Appendix A.5, we also presented experiments on the **multi-span question answering** problem, where the length of an answer varies from one to **dozens of words**. Inspired by the reviewer's comment, we examined the performance of our approach on long-answer questions, that is, questions with an average of **more than 10 words per answer**. As shown in the following tables, our One2Branch considerably outperforms the baseline One2Seq in almost all the settings.
>
> |  w/ doc (extraction task), &#124;ans&#124; > 10 | EM  | PM | EM | PM |
> |  ----  | ----  | ----  | ----  | ----  |
> |  | ***T5-Base*** | | ***T5-Large*** | | |
> | One2Seq  | 29.2 | 61.8 | 22.6 | 60.4  |
> | One2Branch  | **30.1** | **64.4** | **31.2** | **63.1**  |
>
> |  w/o doc (generation task), &#124;ans&#124; > 10| EM  | PM | EM | PM |
> |  ----  | ----  | ----  | ----  | ----  |
> |  | ***T5-Base*** | | ***T5-Large*** | | |
> | One2Seq  | 4.8 | **19.6** | 3.0 | 19.8 |
> | One2Branch | **8.3** | 19.2 | **7.5** | **20.5** |
>
> Excitedly, we appreciate the reviewer's suggestion about other potential applications of our approach. We did not foresee that many opportunities! We will definitely add them to our future work.
>
> *W1. "Firstly, the sota comparisons are focused on pretrained models and sequential decoders. Diverse decoding and generation methods (see some relevant papers below) would also potentially be good for set generation tasks, and it would be interesting to know how branching decoders perform against them."*
>
> Many thanks for providing references about diverse decoding and generation methods. While we would argue that diversity is not necessarily an optimization objective to be pursued in set generation tasks, following the reviewer's idea, we conducted an experiment to compare our One2Branch with Diversity Beam Search (https://www.aaai.org/ocs/index.php/AAAI/AAAI18/paper/view/17329), a representive diverse decoding and generation method mentioned in the review. Due to time limitations, we only completed our experiments on the multi-span question answering problem. As shown in the following tables, our One2Branch outperforms Diversity Beam Search in almost all the settings. More experiments will be added to the camera-ready version.
>
> |  w/ doc (extraction task) | EM  | PM | EM | PM |
> |  ----  | ----  | ----  | ----  | ----  |
> |  | ***T5-Base*** | | ***T5-Large*** | | |
> | One2Seq (Diversity Beam Search)   | 69.1 | **83.1** | 65.3 | 81.2 |
> | One2Branch    | **70.2**  |  82.9 |  **73.3** |  **85.5** |
>
> |  w/o doc (generation task) | EM  | PM | EM | PM |
> |  ----  | ----  | ----  | ----  | ----  |
> |  | ***T5-Base*** | | ***T5-Large*** | | |
> | One2Seq (Diversity Beam Search)   | 14.1 | 34.1 | 15.1 | 35.5 |
> | One2Branch      |  **20.9**  | **37.2**   | **23.3** |**39.7**  |
>
> *Question: "Minor: Also, in section 3.4, why do we need the minimum number of explored paths (k^min) given we are using the threshold-based decoding algorithm?"*
>
> It is used to strengthen the robustness of our decoder. As mentioned in Section 3.4, for each generated sequence we require its **average** token score to be positive, but some token scores can be negative. This is a practical relaxation of our threshold-based decoding. To enable it, we introduce $k^\text{min}$ to keep top-$k^\text{min}$ tokens at each time-step, even if their scores are negative, because they have a chance to utilamtely lead to a sequence having a positive average token score.

---

### Meta-Review · Area_Chair_TeCQ · 2023-12-06

**Metareview:**

This paper addresses set generation by a novel branching decoding algorithm, which comes with a contrastive learning-like ZLPR loss and thresholding during inference.

Reviewers generally agree that the proposed method is interesting, but it's also noticed that the experimentation scope is a little narrow as only keyphrase generation is considered.

**Justification For Why Not Higher Score:**

The experimentation scope is a little narrow as only keyphrase generation is considered.

**Justification For Why Not Lower Score:**

Reviewers generally agree that the proposed method is interesting.

---

### Decision · Program_Chairs · 2024-01-16

Accept (poster)